# Multi-Omic Blood Biomarkers as Dynamic Risk Predictors in Late-Onset Alzheimer’s Disease

**DOI:** 10.3390/ijms25021231

**Published:** 2024-01-19

**Authors:** Oneil G. Bhalala, Rosie Watson, Nawaf Yassi

**Affiliations:** 1Population Health and Immunity Division, The Walter and Eliza Hall Institute of Medical Research, Parkville 3052, Australia; watson.r@wehi.edu.au (R.W.); yassi.n@wehi.edu.au (N.Y.); 2Department of Neurology, Melbourne Brain Centre at The Royal Melbourne Hospital, University of Melbourne, Parkville 3050, Australia; 3Department of Medicine, The Royal Melbourne Hospital, University of Melbourne, Parkville 3050, Australia

**Keywords:** Alzheimer’s disease, genomics, proteomics, metabolomics, risk prediction

## Abstract

Late-onset Alzheimer’s disease is the leading cause of dementia worldwide, accounting for a growing burden of morbidity and mortality. Diagnosing Alzheimer’s disease before symptoms are established is clinically challenging, but would provide therapeutic windows for disease-modifying interventions. Blood biomarkers, including genetics, proteins and metabolites, are emerging as powerful predictors of Alzheimer’s disease at various timepoints within the disease course, including at the preclinical stage. In this review, we discuss recent advances in such blood biomarkers for determining disease risk. We highlight how leveraging polygenic risk scores, based on genome-wide association studies, can help stratify individuals along their risk profile. We summarize studies analyzing protein biomarkers, as well as report on recent proteomic- and metabolomic-based prediction models. Finally, we discuss how a combination of multi-omic blood biomarkers can potentially be used in memory clinics for diagnosis and to assess the dynamic risk an individual has for developing Alzheimer’s disease dementia.

## 1. Introduction

Late-onset Alzheimer’s disease (AD), the leading cause of dementia worldwide, is projected to affect more than 150 million individuals globally by 2050, with the increase mainly driven by population growth and ageing [1]. AD diagnosis is made clinically based on a syndrome of progressive cognitive impairment, typically with a predominant amnestic profile, though the constellation of symptoms can vary considerably. Diagnosis can be supported with the use of biomarkers, such as those derived from biofluids and brain imaging. However, there is significant global variation in the use of these ancillary investigations partly due to the invasiveness of some tests, such as obtaining cerebral spinal fluid (CSF), and the accessibility of advanced imaging including magnetic resonance imaging (MRI) and positron emission tomography (PET) [2]. The clinical interpretation of these biomarkers within different disease stages and in the setting of comorbid cerebral disease also remains challenging. These barriers have thus far prevented the widespread implementation of biomarkers into a unified diagnostic pathway for AD, though there is a movement towards defining and diagnosing AD biologically through the presence of biomarkers [3].

A rapidly expanding body of research has centered around using blood biomarkers to diagnose AD. Blood tests offer a less invasive and more accessible means of assessing biomarkers in patients, especially in the setting of frailty or where geographic or socioeconomic factors prevent access to other biomarker technologies. Genetic analysis using blood samples has also become more widely available due in part to the significant reduction in costs and this has led to the identification of numerous genetic markers associated with AD risk. Technical advances have improved the detection of blood-based proteins and metabolites implicated in AD, such as amyloid-beta (Aβ), tau species, branched-chain amino acids and lipid subspecies [4]. Blood biomarkers have differing temporal profiles, allowing for a more refined characterization of the disease course. They also offer the potential to more accurately predict the risk of an individual developing AD than clinical information alone; thereby, they provide opportunities for earlier diagnoses, improved participant selection for clinical trials and appropriate use of emerging therapies.

In this review, we summarize the recent advances in genomic, proteomic and metabolomic blood biomarkers for AD. We discuss how these types of blood biomarkers can be leveraged to provide a dynamic understanding of AD risk in individuals as they progress through the disease spectrum and respond to treatment. We also highlight how multi-omic blood biomarkers could be incorporated into memory clinics to guide AD diagnosis, especially as revised diagnostic criteria are likely to define AD by the presence of these blood biomarkers [3].

Box | Definitions Related to Large Population Studies

Single-Nucleotide Polymorphisms (SNPs) | Single nucleotides that vary between individuals, with each variation usually occurring at a frequency of >1% in a population. 

Genome-wide Association Studies (GWASs) | Large-scale genome-wide SNP analyses comparing variation frequencies between populations of individuals with the trait of interest and populations of control individuals to statistically quantify associations between the SNPs and the trait risk.

Heritability | Statistical value estimating the proportion of variation seen in a trait that is due to genetic variation between individuals within a population.

Proxy Case | An individual who does not exhibit the trait but has at least one relative (usually first degree) that is affected by the trait of interest and is treated as a case for the purpose of GWASs.

Effect Size | Magnitude of the effect of a specific SNP on the studied trait.

Linkage Disequilibrium | Frequency of association of one allele with another nearby allele in a population, with high linkage disequilibrium indicating an increased observed frequency of association between the two alleles than what would be expected if alleles were associated randomly.

Polygenic Risk Score (PRS) | Risk of trait quantified by sum of effect sizes across multiple trait-associated SNPs.

Receiver Operating Characteristic (ROC) Curve | Curve generated by plotting the true positive rate versus the false positive rate of a diagnostic model. 

Area under the ROC Curve (AUC) | The calculated area represents the probability that a diagnostic model provides a higher numerical value for a random positive case than a random negative case.

## 2. Genetic Markers as a Measure of Baseline Risk

The genetic contribution to AD is substantial with heritability (see Box) estimated to be between 60% and 80% based on twin-studies [5]. Early linkage studies in familial Alzheimer’s disease identified a strong association with the apoliprotein E (*APOE*) ε4 allele on chromosome 19 [6,7,8]. While this is the strongest associated locus for AD, accounting for ~5% of the heritability [9] and a three- to four-fold increased risk [10], it is clear that the *APOE* ε4 is not the only genetic contributor as roughly 50% of individuals with AD do not carry the ε4 allele [11]. Moreover, the population frequency of the *APOE* ε4 allele ranges from 0.05 to 0.30, decreasing with age but varying considerably by region and ancestry [12,13,14,15]. These findings demonstrate the need to discern the ‘missing heritability’ of AD.

### 2.1. Genome-Wide Association Studies and Polygenic Risk Scores

Genome-wide association studies (GWASs) have allowed the identification of single nucleotide polymorphisms (SNPs) associated with AD risk using case-control cohort studies. While the initial GWAS, with only a few thousand cases and controls, confirmed the association of the *APOE* locus [16], recent studies with substantially larger and overlapping cohorts have implicated significantly more risk loci (Table 1) [17,18,19,20,21]. New risk loci have been elucidated due to meta-analyses of previous large GWASs as well as including ‘proxy’ cases, in which individuals who are otherwise cognitively unaffected are considered AD cases if either parent had AD [22]. In this way, an AD GWAS meta-analysis of 111,326 clinical and ‘proxy’ AD cases and 677,663 controls of predominantly European ancestry found 75 risk loci, of which 42 were new compared to prior analyses [17]. Interestingly, the genes associated with these risk loci cluster to cellular pathways related to Aβ, tau, lipids, endocytosis and immunity, in line with the current understanding of AD neuropathology.

In addition to generating hypotheses about implicated genes and pathways in AD [17,23], GWAS results can be leveraged to stratify individuals along their AD risk based on their combination of SNPs through the use of polygenic risk scores (PRSs) [24]. The objective of PRSs is to predict an individual’s predisposition for a certain disease, such as AD, by summing effect sizes, also known as odds ratios (ORs), of risk-associated SNPs [25]. Therefore, the PRSs numerically represent a baseline risk of developing AD based solely on an individual’s combination of genetic variants (Figure 1) [26]. An AD-PRS based on 83 SNPs derived from the GWAS meta-analysis described above [17] is significantly associated with risk of incident AD in prospective population-based cohorts (hazard ratio [HR] = 1.93 [1.75–2.13 95% confidence interval]) as well as with the risk of conversion from mild cognitive impairment (MCI) to AD over time (HR = 1.63 [1.42–1.87]). Importantly, the AD-PRS risk is additive to that of age and *APOE* status, demonstrating how risk prediction can be improved with the inclusion of polygenic information. In a targeted study of over 2000 Swedish individuals over 70 years old, a 39-SNP AD-PRS was associated with incident dementia in individuals who were *APOE* ε4 non-carriers (HR = 1.22 [1.10–1.35], *p* = 2 × 10^−4^) [27]. With respect to *APOE* ε4 homozygosity, the AD-PRS was significantly higher in individuals with young-onset Alzheimer’s disease (onset before 65 years old) compared to individuals who were cognitively unimpaired and older than 75 years (OR = 8.39 [2.0–35.2], *p* = 0.003) [28]. These findings further support the previous observations that a substantial proportion of the genetic risk for AD is derived from outside the *APOE* locus.

Information from GWASs can also be vital in designing clinical trials. Utilizing GWAS data to construct a polygenic hazard score (PHS) enables prediction of the lifetime trajectory of AD risk, even during the preclinical state (defined as having evidence of AD neuropathology in the absence of symptoms or cognitive impairment), by indicating an individual’s instantaneous risk of developing AD [25,29]. In a European-based cohort study, non-*APOE* ε4 carriers with a PHS in the tenth decile had an AD age-of-onset 10 years earlier than those in the first decile [30]. These findings were replicated in another European-based cohort study, where those individuals in the top 5% for PHS had on average a 20-year earlier age-of-onset compared to those in the bottom 5% for PHS [31]. In a prospective study of asymptomatic Europeans, individuals in the sixth sextile for PHS developed AD roughly 8 years earlier than those in the first sextile [32]. Furthermore, individuals with MCI in the tenth decile of risk (based on AD-PRS, *APOE*, age and sex) had nearly a five-fold higher rate of conversion to AD than those in the lowest decile [33]. These studies demonstrate how GWAS data can inform genetic-based risk scores to predict instantaneous risk and to estimate age-of-onset for AD. 

### 2.2. Clinical Applications of Polygenic Risk Scores

Utilizing polygenic scores may help design more efficient clinical trials in AD, particularly by overcoming the slow rates of cognitive decline and time-limited study windows constraining clinical trials [34]. AD clinical trials usually enroll genetically heterogenous participants with genetic-based analyses typically performed in a post-hoc manner, especially for the *APOE* genotype [35,36,37]. Such heterogeneity within the trial may have the unintended consequence of confounding the therapeutic effects, as seen in Parkinson’s disease [38]. Therefore, using polygenic scores to identify appropriate study participants can help reduce the size and costs of clinical trials [39]. For example, by dichotomizing individuals who have MCI at baseline using an AD-specific PHS30, clinical studies may require up to 60% percent less participants by 3 years when they are limited to participants with the 50% highest PHS risk [40]. However, when such a strategy is applied to individuals with normal cognition at baseline, it is estimated that over 3 years are needed to observe a significant therapeutic effect. Moreover, when a PRS is applied to patients with established AD, it can only account for a small amount of the variability seen in cognitive decline, suggesting that the applicability of an AD-PRS in clinical trials depends heavily on where the study population sits along the AD continuum [41].

The interaction between lifestyle and genetic factors affects the timing and risk of developing AD [42,43]. While lifestyle changes alone will not completely eliminate AD risk, they can potentially, partially mitigate the burden of having a high genetic risk. In a prospective cohort study of individuals with European ancestry from the Rotterdam Study, cognitively unimpaired individuals with a low to intermediate AD-PRS had a lower risk of dementia when coupled with a favorable set of lifestyle factors (such as alcohol abstinence, non-diabetic, healthy diet, social engagement and physical activity), compared to an unfavorable profile of lifestyle factors [44]. However, in this study, the converse was not true as a favorable lifestyle profile did not reduce the risk of cognitively unimpaired individuals with a high score for AD-PRS. Contrastingly, in a separate retrospective study of over 500,000 individuals from the UK Biobank, a high AD-PRS was ameliorated with a healthy lifestyle compared to an unhealthy one (OR = 0.68 [0.51–0.90], *p* = 0.008) [45]. Of note, this study demonstrated that lifestyle factors did not vary considerably based on the genetic risk, indicating that lifestyle factors and behaviors are not merely a proxy for the genetic predisposition of AD but have independent associations with AD risk. When looking at diet alone, a strong adherence to a Mediterranean diet is associated with a lower dementia risk, irrespective of polygenic risk scores [46], though the association between diet and cognitive decline is unclear [47]. These studies demonstrate emerging evidence, at times conflicting, of the interaction between genetic and lifestyle risk factors for AD. More work is needed in this area to further elucidate these relationships.

The cost-effectiveness of generating and utilizing an AD-PRS has not been well studied, partly due to the uncertainty of when to apply the PRS in individuals given the long duration between the preclinical and symptomatic stages of AD. However, the potential benefits of an AD-PRS can be extrapolated from studies demonstrating improved resource allocation when a disease-specific PRS is applied to cardiovascular disease [48,49], and breast cancer [50]. Evidence of behavior change based on knowledge of one’s PRS is seen with cardiovascular disease, where a healthy behavior change, albeit slight, is more likely to be undertaken in those who knowingly have a high genetic risk (OR = 1.10 [1.03–1.17, *p* = 0.044) [51]. A total of 97% of participants in this study believed that genetic factors influenced disease risk, providing optimism for uptake by individuals before symptom onset.

Caution is needed when applying PRSs to diverse populations, as the PRSs are mostly derived from studies with an over-representation of individuals with European ancestry, potentially limiting their applicability to non-European cohorts and increasing health disparities in already under-resourced communities [52]. Large cohort studies such as BioBank Japan [53], H3Africa [54] and China Kadoorie Biobank [55] are providing a broader base for developing more equitable and applicable PRSs for various diseases. Supporting a need for these diverse biobanks is a study that identified two novel loci not identified in prior European-centric GWAS analyses by performing a multi-ancestry (of East Asian, African American, Caribbean Hispanic populations and European) meta-analysis using existing AD GWAS databases [56]. Bioinformatic methods are also being developed to effectively apply existing PRSs derived from one ancestral group (such as European) to more diverse study populations [57,58]. Using such techniques, an AD-PRS derived from individuals with European ancestry demonstrated improved prediction in a Korean population (OR = 1.95 [1.40–2.72], *p* < 0.001) [59]. These different lines of research will help improve the transferability of existing PRSs, thereby addressing some of the research-based societal inequalities inherent in genetic studies.

These types of studies clearly demonstrate that genetic information can be used to develop a PRS to represent an individual’s genetic risk of developing AD over their lifetime. This has the advantage of potentially identifying those at the highest genetic risk as early as birth as this is a static risk. Targeted intervention and resources, such as lifestyle measures and earlier cognitive screening programs, can be considered in such individuals to minimize the rate of cognitive decline and potentially abort disease progress. However, this has important ethical and societal ramifications as the above genetic studies have also shown that not all individuals with a high PRS develop AD (Figure 1). Therefore, more dynamic and real-time blood biomarkers are needed to better stratify an individual’s contemporaneous risk of developing AD.

## 3. Blood-Based Protein Biomarkers as a Measure of Dynamic Risk

While considerable progress has been made in uncovering the genetic architecture of AD through large GWASs, the resulting PRSs represent a static risk that is embedded within an individual’s genetics. The analysis of more dynamic biomarkers may provide further insights into the real-time risk of AD at a particular stage of life (Figure 1). Technological improvements in biomarker detection have led to improved utility for AD diagnosis. In particular, ultrasensitive detection is possible with the use of single molecule array (Simoa) and mass spectrometry, allowing for the quantification of ultralow concentrations (pico- and femtomolar ranges) [4,60,61]. Blood levels of AD-implicated proteins such as neurofilament light chain (NfL), Aβ and tau (including hyperphosphorylated species) demonstrate strong concordance with CSF levels [62,63,64], thereby increasing their utility in diagnosing AD. Potential applications of different protein blood biomarkers in assessing AD risk are discussed below.

### 3.1. Neurofilament Light Chain

NfL is a cytoskeletal protein found predominantly in neuronal axons with a role in axonal growth and stability, with CSF and blood levels increasing after axonal damage [65]. Elevated plasma NfL levels are detected in a wide range of neurodegenerative diseases [66] including AD [67,68], frontotemporal dementia (FTD) [69], amyotrophic lateral sclerosis [70] and HIV-associated dementia [71]. However, NfL can also be elevated in acute non-neurodegenerative causes of brain injury including stroke [72] and encephalitis [73], highlighting that clinical context is important for biomarker interpretation. The role of NfL may be in discriminating between neurological disorders and psychiatric disorders, both of which can present with cognitive and memory dysfunction, a common clinical dilemma [74,75].

The ‘real world’ utility of blood NfL levels for assessing AD risk has been studied in memory clinics. In one prospective study of over 100 patients assessed in a tertiary memory clinic, physicians found knowledge of serum NfL levels diagnostically useful in patients under 62 years of age (60% useful) and in male patients (62% useful), as well as in those with a diagnostic uncertainty (67% vs. 51% useful in those patients with no diagnostic uncertainty) [76]. These findings were independent of knowledge of CSF NfL levels, suggesting that there is growing confidence in the use of blood NfL levels amongst physicians. Analysis of plasma NfL levels in over 550 patients with established diagnoses in a retrospective memory clinic study also found higher values in those with neurodegenerative conditions compared to non-neurodegenerative conditions [77]. Moreover, the increase in NfL levels correlates with the degree of cognitive impairment (higher levels seen in dementia compared to those with MCI, which was in turn higher than those with subjective cognitive impairment) [77,78,79]. However, the addition of plasma NfL levels to a diagnostic model based on clinical factors (age, cognitive test scores and APOE status) does not significantly increase the area under the receiver operating characteristic curve (AUC, 0.83 [0.78–0.87] versus 0.81 [0.77–0.85] for a model without NfL levels) for discriminating neurodegenerative from non-neurodegenerative conditions in patients with established cognitive impairment, suggesting that knowledge of blood NfL level may not be as relevant once substantial symptoms have developed [77]. Further, longitudinal blood NfL levels are not predictive of conversion to AD [80,81].

While NfL is emerging as an attractive biomarker for neurodegenerative conditions, its low specificity reduces its ability to serve as an AD-specific biomarker [4,82]. Moreover, cut-off values have not been well established for clinical use, though NfL levels increase in an age-dependent manner [82,83] and age-adjusted models have been proposed [84]. Importantly, the effects of ancestry on normative values for NfL are unclear with conflicting results [75,85,86,87]. These open questions need to be addressed before NfL can be used robustly in a clinical setting to identify neurodegeneration [88].

### 3.2. Amyloid-Beta

Accumulation of Aβ plaques is a key component of AD pathology, with levels of Aβ1-42 changing decades before symptom onset. While a decrease in Aβ1-42 levels within the CSF has been robustly associated with AD [89], the association of blood levels was less clear due to concentrations being up to 100-fold lower in blood compared to CSF. Early meta-analyses of blood Aβ1-42 levels measured using plate-based immunoassays such as ELISA did not find significant differences between AD and healthy controls [90]. Measurements using mass spectrometry demonstrate a stronger correlation between plasma and CSF Aβ1-42 levels [91,92], suggesting that earlier equivocal results for Aβ likely reflect technological challenges rather than true pathobiology. Still, plasma Aβ levels and robustness differ significantly based on the assay used, limiting widespread uptake as a useful clinical blood biomarker [91].

To assess how plasma Aβ1-42 may serve as a biomarker for AD risk, a subgroup analysis of the Rotterdam study of over 450 older individuals (mean age of 68 years) found that lower plasma Aβ1-42 levels were associated with increased dementia incidence (HR = 1.27 [1.02–1.58]), especially among those individuals that were non-APOE ε4 carriers (HR = 1.47 [1.09–1.99]) [93]. In older individuals with subjective cognitive concerns but without a dementia diagnosis on the initial visit, a lower plasma Aβ42/40 ratio (value of Aβ1-42/Aβ1-40) demonstrated a steeper cognitive decline over a median follow-up of 3.9 years compared to those individuals with a higher Aβ42/40 ratio [94,95]. A lower plasma Aβ42/40 ratio is also seen in individuals with MCI who develop dementia compared to those who do not [96]. Moreover, individuals with the highest Aβ42/40 ratio have a significantly lower dementia risk (HR 0.52 [0.31–0.86]) over a 3-year period. Of note, plasma levels of Aβ (either as Aβ1-42, Aβ1-40 or Aβ42/40 ratio) do not significantly differ along the AD continuum (from cognitively unimpaired Aβ+ individuals to MCI to AD), indicating that it may not be useful in prognosticating disease progression [81,97].

As discussed above, Aβ deposition within the brain starts well before symptom onset. Detecting deposition non-invasively in vivo is possible with Aβ-PET emerging as a powerful imaging tool [98]. While diagnostically sensitive, Aβ-PET is resource intensive and not easily accessible in many countries. Plasma Aβ1-42 levels show excellent performance characteristics with high AUC values (above 0.9) in predicting Aβ-PET levels [99,100]. Consequently, new and efficient investigation pathways can be developed for individuals suspected for AD. For example, by using plasma Aβ to screen individuals with cognitive concerns and only proceeding to a Aβ-PET scan if plasma levels are abnormal, there would be over a 50% reduction in the number of PET scans needed to diagnose AD via PET imaging [101,102]. These findings highlight the potential role of blood Aβ measurements as a population-screening tool for AD, especially in those populations with a lower prevalence of AD.

Aβ deposition is not restricted to AD and is detected in non-AD causes of dementia, including dementia with Lewy bodies (DLB), Parkinson’s disease dementia (PDD), FTD and vascular dementia (VaD). In these non-AD cases, Aβ levels, as measured by Aβ-PET, also increase with age and APOE ε4 carrier status but vary in cortical distribution relative to the underlying dementia diagnosis [103,104]. With respect to blood biomarkers, plasma Aβ levels also vary amongst non-AD dementia types and may be higher in VaD compared to AD [105,106,107,108]. However, it is not well established how accurately blood Aβ levels, either as Aβ1-42 or as Aβ42/40, can discriminate between dementia subtypes early in the disease process and studies are needed to further investigate this.

### 3.3. Tau

Tau tangles, like Aβ, are a quintessential feature of AD, with tau-hyperphosphorylation leading to significant pathology [109]. Total-tau (t-tau) increases in the CSF and plasma following various causes of neuronal injury such as ischemic stroke and cardiac injury, as well as in neurodegenerative conditions including AD, DLB and FTD. The non-specific nature of elevated t-tau limits its ability to discriminate between AD and non-AD dementia. 

In contrast to CSF, blood t-tau levels reflect production from the central nervous system (CNS) as well as from peripheral tissues (such as liver, heart and kidney), explaining why blood t-tau levels are not considered diagnostic. Given that only one-fifth of plasma t-tau originates from the CNS [110], assays that specifically measure brain-derived tau levels are needed. By exploiting the fact that peripherally derived tau contains exon 4a, which is not found in CNS-derived tau109, a unique tau antibody has been developed to specifically measure plasma levels of brain-derived t-tau (BD-tau) [111]. Using this antibody, BD-tau levels correlate with CSF t-tau levels and are able to differentiate between autopsy-confirmed AD vs. non-AD cases (AUC = 0.86 [0.76–0.97]). When tested in memory clinics, BD-tau analysis is able to differentiate AD from non-AD neurodegenerative causes with AUC ranging from 0.78 (for progressive supranuclear palsy) to 0.99 (for the agrammatic variant of primary progressive aphasia due to a progranulin mutation).

In addition to t-tau and BD-tau, there are nearly 100 known post-translational modifications of tau [109]. Some of the tau species are phosphorylated at unique threonine sites (p-tau) and have been found to be highly specific for AD [112,113]. The role of these p-tau species in AD prediction is highlighted below.

#### 3.3.1. P-Tau181

Tau phosphorylated at threonine 181 (p-tau181) is one of the most studied tau species in AD. Using mass spectrometry and ultra-sensitive immunoassays, blood p-tau181 levels can differentiate between cognitively unimpaired individuals and those that have MCI or AD. In a prospective cohort study of 589 individuals from the Swedish BioFINDER cohort, plasma p-tau181 levels were strongly correlated with CSF p-tau181 levels, Aβ-PET and tau-PET [114]. Plasma p-tau181 was elevated in preclinical AD cases (individuals who were cognitively normal but with Aβ-PET positivity), and was able to discriminate between AD and non-AD dementia cases (AUC = 0.94 [0.90–0.99]). Similarly, in a retrospective North American cohort study of over 400 individuals, plasma p-tau181 levels were 3.5-fold higher in AD than cognitively unimpaired individuals and successfully discriminated between both clinically diagnosed and autopsy-confirmed AD and FTD cases (AUC = 0.87–0.89) [115]. The discriminatory power of blood p-tau181 was again demonstrated in a UK cohort of [115] individuals (AUC = 0.97 [0.94–1.00], autopsy-confirmed AD vs. non-AD dementia) [116] as well in separate North American and Swedish cohorts, with an AUC = 0.83–1.00 for AD vs. FTD and AUC = 0.92 for clinically diagnosed AD vs. VaD [117]. Importantly, p-tau181 tracks along the AD continuum (as measured by CSF Aβ levels and Aβ-PET load) and with cognitive decline, further supporting its role as a dynamic AD risk marker [81,97].

The robustness of plasma p-tau181 across ancestries was demonstrated by a prospective Spanish cohort study of 349 individuals (AUC = 0.96 for clinically diagnosed AD vs. cognitively unimpaired individuals) [118] and in a small Thai cohort study of 51 individuals (AUC = 0.84 [0.73–0.94]) [119]. However, p-tau181 performance was reduced among non-Hispanic White Americans (AUC = 0.69 [0.59–0.80]) and Black Americans (AUC = 0.63 [0.51–0.74]), and was considerably lower in Hispanic Americans (AUC = 0.51 [0.40–0.64]) [120]. More studies are needed to assess the generalizability of plasma p-tau181 amongst patients with different ancestral backgrounds.

#### 3.3.2. P-Tau217

Other phosphorylated tau species are being investigated for their ability to discriminate between AD and non-AD dementia along the disease continuum. One such species is p-tau217 (phosphorylated at threonine 217), which has demonstrated improved performance in CSF, compared to CSF p-tau181, in distinguishing clinically diagnosed AD from non-AD dementia [121,122]. Similarly, plasma p-tau217 has also been found to better discriminate between autopsy-confirmed AD and non-AD cases (AUC = 0.89 [0.81–0.97]) compared to plasma p-tau181 (AUC = 0.72 [0.60–0.84]) [123]. Similarly, p-tau217 was shown to be superior to p-tau181 in discriminating between clinically diagnosed AD and non-AD cases (p-tau217 AUC = 0.96 [0.93–0.98], p-tau181 AUC = 0.81 [0.74–0.87], *p* < 0.001) as well as being more specific for Aβ-PET positivity than p-tau181 [110]. The specificity of p-tau217 for AD compared to other neurodegenerative tauopathies is further supported by a North American multicohort study where p-tau217 differentiated clinically diagnosed AD from FTD, with an AUC = 0.93 (0.91–0.96), compared to an AUC = 0.91 (0.88–0.94) for p-tau181 (*p* = 0.01) [124]. With respect to p-tau performances amongst individuals from different ancestral backgrounds, similar to p-tau181, plasma p-tau217 performed better in non-Hispanic White Americans (AUC = 0.71 [0.61–0.82]) and Black Americans (AUC = 0.68 [0.57–0.78]), but had a poor performance in Hispanic Americans (AUC = 0.52 [0.40–0.64]), considerably lower accuracy than those seen in European studies [120].

With respect to PET imaging, plasma p-tau217 levels can distinguish cognitively unimpaired individuals who are Aβ-PET positive from Aβ-PET negative cognitively unimpaired individuals even when the former individuals’ tau-PET scans are negative in the entorhinal cortex, a region involved early in AD-related tau pathology [125]. Furthermore, individuals with negative tau-PET imaging within the entorhinal cortex demonstrate a 2.2% increase per year of tracer uptake if they have higher baseline levels of plasma p-tau217. Interestingly, plasma p-tau217 levels in PSEN1 E280A mutation carriers (a form of autosomal dominant Alzheimer’s disease) are significantly altered at 24.9 years of age compared to non-carriers, which is approximately 20 years earlier than the expected age for symptom onset within the PSEN1 mutation carrier population [123]. This finding has been replicated in another cohort of cognitively unimpaired PSEN1 E280A mutation carriers, where plasma p-tau217 levels are higher compared to non-carriers and predict a higher burden of Aβ and tau as measured using PET imaging [126].

P-tau217 is also associated with AD progression risk. In cognitively unimpaired individuals who are Aβ-PET positive, a higher baseline plasma p-tau217 level confers an increased risk of conversion to clinically defined AD over a median follow-up time of 6 years (HR = 2.03 [1.57–2.63], *p* < 0.001]) [127]. Longitudinal plasma p-tau217 measurements demonstrate a greater increase in individuals with MCI who convert to AD compared to those who did not [128]. These studies demonstrate the potential utility of p-tau217 in identifying individuals at risk for AD well before symptoms and PET imaging changes emerge.

#### 3.3.3. P-Tau231

P-tau231 (phosphorylated at threonine 231) is emerging as another potential blood biomarker which is highly sensitive for AD. Using a newly developed ultra-sensitive Simoa assay, p-tau231 was detected in all clinical stages of AD, including in individuals with MCI and sub-threshold signals in Aβ-PET, with levels increasing alongside disease progression [129]. Plasma p-tau231 is also seen in the early stages of AD and able to differentiate between Braak 0 (no deposition of Aβ plaque) and Braak I-II (Aβ plaque confined to the transentorhinal region) stages, which has not been observed with p-tau181 [129,130]. However, plasma p-tau231 is similar to p-tau181 in differentiating between AD and non-AD neurodegenerative cases, including limbic age-related TDP-43 encephalopathy and hippocampal sclerosis, both of which can clinically present similarly to AD [130,131]. Further studies are needed to elucidate the role of p-tau231 in dynamic AD risk profiling and how it differs from the other p-tau species.

These studies highlight that novel blood tau antibody development, including different assay systems that target known p-tau species [132], can further improve the utility of p-tau and brain-specific tau as sensitive blood biomarkers for AD prediction.

### 3.4. YKL-40

The role of neuroinflammation in AD is becoming increasingly recognized [133]. YKL-40, also known as chitinase 3-like protein 1 (CHI3L1), is a highly conserved acute-phase glycoprotein involved in inflammation-activated remodeling and may be an indicator of neuroinflammation, but its exact function within the brain is not completely understood [134]. Interest in YKL-40 as a biomarker for AD was generated by early studies that found higher CSF YKL-40 levels in those with MCI and mild AD compared to cognitively unimpaired individuals; these levels also correlated well with CSF Aβ and p-tau181 [135]. Similarly, blood YKL-40 levels are higher in individuals with AD compared to healthy control, with levels increasing with disease severity [136,137,138,139]. Interestingly, CHI3L1-associated SNPs correlated with blood protein levels and AD risk in a Han Chinese population [136]. Blood YKL-40 levels are also negatively correlated with structural (regional volume and cortical thickness) MRI changes in individuals with AD, but not with cognitive decline, suggesting it may serve as a generic marker of neurodegeneration [138,140,141]. The exact association of blood YKL-40 and AD needs further study as prospective analyses in cognitively unimpaired individuals suggest higher levels may be potentially protective (with reduced Aβ accumulation and improved cognitive testing) [142], while other studies suggest YKL-40 as a detrimental marker [143,144] and possibly specific to certain ethnic groups [145].

### 3.5. Soluble Triggering Receptor Expressed on Myeloid Cells 2

Triggering receptor expressed on myeloid cells 2 (TREM2) is a transmembrane receptor expressed in many immune-related cells, including CNS microglia [146]. Rare SNPs in this gene are second only to APOE in terms of magnitude of associated genetic risk of AD, with the R47H variant of TREM2 increasing AD risk two- to three-fold [147,148]. While there has been immense interest in TREM2 with respect to the pathogenesis of AD, its use as a peripheral biomarker has been recently proposed given the finding of an increase in blood TREM2 expression in AD patients compared to those cognitively unimpaired [149,150]. However, the association of AD status with levels of the soluble form of TREM2 (sTREM2) within the CSF and blood is inconsistent, possibly reflecting technological challenges in detecting this protein [151,152,153,154,155]. Considerably more work is needed to determine if TREM2 and/or sTREM2 are robust biomarkers for AD risk.

### 3.6. Glial Fibrillary Acidic Protein

Glial fibrillary acidic protein (GFAP) is an abundant intermediate filament cytoskeletal protein highly expressed in astrocytes, with a role in neuro-inflammation and astrocyte reactivity. As such, GFAP’s use as an AD blood biomarker is promising as higher levels are found in individuals with AD compared to those cognitively unimpaired [156,157] and MCI [158]. Moreover, plasma GFAP levels predict conversion to AD in individuals with MCI over a 5 year period, independent of APOE ε4 status and age (AUC = 0.84 [0.77–0.91]) [159]. Plasma GFAP levels also correlate with Aβ-PET positivity [160,161], but not tau-PET [162,163], and can more accurately reflect CNS Aβ levels than other markers of inflammation such as YKL-40 or sTREM2 [162]. Interestingly, plasma GFAP may discriminate Aβ-PET positivity better than CSF GFAP (plasma AUC = 0.69–0.86 vs. CSF AUC = 0.59–0.76), as well as demonstrating a higher magnitude of change along the AD continuum [164]. Of note, a rise in plasma GFAP levels is also seen in Lewy body dementia [157] and FTD [165], suggesting that plasma GFAP levels are reflective of the reactive astrogliosis occurring in neurodegeneration more broadly. Nonetheless, the growing importance of GFAP in AD risk is evidenced by its potential inclusion in revised diagnostic criteria for AD [3].

### 3.7. Comparing Blood-Based Protein Biomarkers as Risk Predictors

As blood biomarkers demonstrate improved diagnostic performance, a natural question arises as to which of these protein biomarker(s) is/are most powerful in identifying AD risk. Recent studies have compared the various biomarkers at different stages along the AD continuum (Table 2). With the increasing rate of publications in this field, publicly available databases such as AlzBiomarker [166] provide updated biomarker meta-analyses, allowing for comparisons of multiple proteins of interest.

In a study of over 300 individuals of European background, plasma p-tau181 outperformed other blood biomarkers (GFAP, NfL, t-tau and Aβ42/40) when distinguishing between clinically diagnosed AD and cognitively unimpaired individuals (AUC = 0.91 [0.86–0.96] vs. AUC = 0.67–0.82 for other blood biomarkers) as well between individuals with MCI who converted to AD and those who did not (AUC = 0.77 [0.61–0.84] vs. AUC = 0.60–0.67 for other blood biomarkers) [158]. Interestingly, combining p-tau181 with the other biomarkers did not increase the diagnostic accuracy in this study. Contrastingly, three separate observational cross-sectional studies totaling over 800 individuals with North American and European backgrounds found no significant difference in diagnostic accuracy of AD between p-tau181 (AUC 0.67–0.87) and GFAP (AUC 0.69–0.86) [164]. In preclinical AD (defined as Aβ-PET positive with normal cognitive profiles), GFAP (AUC = 0.79 [0.69–0.89]) was also not statistically different to p-tau181 (AUC = 0.74 [0.63–0.85]) in discriminating against Aβ-PET negative cognitively unimpaired individuals [167]. In a prospective study of over 110 Swedish and North American preclinical AD individuals, plasma p-tau217 was superior to p-tau181, p-tau231 and GFAP in predicting cognitive decline [127].

Predicting levels of amyloid burden, a risk factor for AD development, is important to facilitate screening of preclinical AD individuals. Plasma p-tau217 and p-tau231 are emerging as potential blood biomarkers for detecting low but abnormal levels of amyloid burden (as quantified by Aβ-PET), compared to p-tau181, GFAP and NfL. The 168 Plasma p-tau231 levels are abnormal at a significantly lower Aβ-PET Centiloids (26.4) than p-tau217 (35.4 Centiloids). The Centiloid scale is a standardized metric for the amyloid signal in Aβ-PET imaging, with 30 Centiloids considered to a cut-off for Aβ-PET positivity. Both p-tau217 and p-tau231 also demonstrate the strongest association with disease progression (compared to p-tau181, GFAP and NfL) [168]. Interestingly, p-tau231 is significantly elevated at lower thresholds of Aβ-PET Centiloids compared to p-tau217, p-tau181, GFAP and NfL [169]. However, it is p-tau217, and not p-tau231, that demonstrates longitudinal changes (over 4 years) that are correlated with AD progression. These findings suggest that p-tau231 may be useful in identifying at-risk individuals for AD early in the disease process, while p-tau217 is useful in tracking disease progression, both of which have implications for dynamic AD risk profiling.

Other combinations of blood biomarkers demonstrate limited utility. In cognitively unimpaired individuals, a biomarker combination of plasma p-tau181 and p-tau217 along with the APOE genotype is not significantly better at predicting conversion to AD (AUC = 0.88 [0.82–0.95]) compared to p-tau181 alone (AUC = 0.84 [0.77–0.92]) [170]. In individuals with MCI, a combination of p-tau181, p-tau217 and Aβ42/40 ratio more accurately predicts AD conversion (AUC = 0.87 [0.82–0.92]) compared to that of any single biomarker, but it was not statistically different than a model with five blood biomarkers (p-tau181, p-tau217, Aβ42/40, NfL and APOE ε4 status) with an AUC = 0.89 [0.85–0.93] (*p* = 0.10) [170]. A combination of p-tau217 levels and Aβ42/40 ratio measured in antemortem plasma strongly predicts AD amyloid and tau load in postmortem analysis with an AUC = 0.89 [0.82–0.96], but not significantly better than p-tau217 alone (*p* = 0.124) [171]. While further studies are needed, these findings suggest that simply testing more biomarkers will not necessarily improve diagnostic accuracy. Instead, in order to maximize the predictive power of the test, selecting the appropriate biomarker(s) may depend on the stage along the AD continuum at which an individual is being assessed [172].

### 3.8. Dynamic Changes in Blood-Based Protein Biomarkers in Response to Anti-Amyloid Therapy

The emergence of anti-amyloid therapies is making disease modification in AD a possibility. Three monoclonal antibodies (aducanumab, lecanemab and donanemab) targeting various Aβ species demonstrate a slowing in progression in early clinical stages of AD, with lecanemab demonstrating up to a 35% reduction in the rate of cognitive decline [173,174,175]. While the main biomarkers used in these studies included Aβ-PET, plasma blood biomarkers were used as exploratory endpoints and showed dynamic changes in response to anti-amyloid therapies. For example, a steady increase in plasma Aβ42/40 ratio and decrease in p-tau181 and GFAP were observed with lecanemab treatment compared to placebo; a lesser magnitude of change was observed with plasma NfL [175]. With donanemab, there was nearly a one-third reduction in levels of p-tau217 and a one-sixth reduction in levels of GFAP compared to placebo [174,176]. Similarly, high-dose aducanumab led to a 20% reduction in levels of p-tau181 compared to placebo [173]. While these studies have demonstrated how blood biomarkers can be used to monitor for responses to anti-amyloid therapy (Figure 1), it is unclear if blood biomarkers further change (in either direction) after therapy is discontinued, indicating remission or progression of AD pathology, and how these changes correspond to AD symptom progression. Nonetheless, the new anti-amyloid therapies are facilitating a paradigm shift in how blood biomarkers are helping to understand AD pathology and treatment. Such a shift in the utilization of blood biomarkers may also shed light on the effects of existing treatments, such as cholinesterase inhibitors and memantine, as well as molecules currently under investigation—an area of research where insufficient data exist [177,178].

## 4. The Metabolomic Profile as a Dynamic Risk Predictor

Age, sex and APOE ε4 genotype are not only some of the strongest risk factors for AD, but also have a significant role in metabolism [179], thus, implicating metabolic pathways in AD [180]. Neuropathological examination of brains with AD demonstrates metabolite dysregulation, indicating a potential role for the metabolome as a risk marker for AD [181]. Large population studies show that metabolic profiles are able to improve prediction models for all-cause dementia [182] and that profiles assessed at midlife correlate with dementia risk over the subsequent 20 years [183].

An emerging area of research is whether metabolite profiles can be leveraged to distinguish AD from other conditions. A panel of 11 metabolites (Table 3) measured in blood was able to identify individuals with AD from non-AD causes of dementia (PDD, FTD, DLB and VaD) as well as from cognitively unimpaired individuals in a replication cohort of over 300 Chinese individuals, with an AUC of 0.96–0.97 (*p* < 0.001) [184]. While each of the 11 metabolites was differentially distributed between AD and cognitively unimpaired individuals (*p* < 0.001), the diagnostic ability of each metabolite was much lower than the 11 metabolite panel, with the former having AUCs between 0.63 and 0.73. Similarly, an 8 metabolite index score, derived using machine-learning, was able to discriminate between individuals with MCI that progress to AD and those that remained stable over a 3- to 5-year period [185]. Further evidence of the improved AD risk prognostication was provided through a pilot study that demonstrated increased diagnostic accuracy for MCI and AD using a model incorporating a broad metabolomic panel (AUC = 0.95) compared to baseline models consisting of only age, sex and APOE ε4 status (AUC = 0.72–0.79) [186].

Branched-chain amino acids (leucine, isoleucine and valine) are associated with AD risk, with lower blood levels of circulating branched-chain amino acids corresponding to higher AD risk [187,188,189]. Interestingly, the blood levels of these amino acids are largely determined by dietary intake, so the biological significance of lower levels seen in AD is unclear and may represent a nutritional and/or metabolic impact on AD. Lipid species profiles have also been interrogated, given the altered metabolism and lipid-associated GWAS loci noted in individuals with AD [17,22,190]. A meta-analysis of metabolites from 1912 individuals found that an addition of 10 species representative of different lipid clusters to a prediction model consisting of age, sex, body mass index and APOE ε4 status improved the diagnostic accuracy for AD cases compared to controls [191]. Similarly, specific lipid subspecies predicted risk of progression in individuals with MCI compared to preclinical AD individuals (with increased CSF p-tau/Aβ1-42 ratio), even when adjusted for APOE ε4 status (HR = 1.97–1.99) [192]. Interestingly, these effects are sex-dependent with different lipid subspecies profiles predicting progression risk according to sex. Other subclasses of metabolites, such as acylcarnitines [193] and sphingolipids [194], are also associated with AD risk and conversion from MCI, but further work is needed to clarify these findings.

Metabolites may reflect more subtle changes due to AD pathology compared to protein biomarkers as the former represents the combined influence of genetic, protein and environmental factors and are more adept at crossing the blood–brain barrier. However, given the complexities of the mass spectrometry technology needed to measure metabolites, resulting in heterogenous results, the regular use of metabolites as part of AD work-up needs further validation and feasibility studies [195]. Metabolites may also have a role in assessing participants in clinical trials, especially for response to treatments, but research in this area is lacking.

## 5. Large-Scale Proteomic Studies Predict Alzheimer’s Disease Risk

As discussed above, specific proteins such as p-tau species and Aβ provide strong predictive power for AD-risk. Soley focusing on the blood levels of these proteins may confound how the response to AD therapies, such as with anti-amyloid and potentially anti-tau antibodies, is interpreted; ideally, blood biomarkers that are not targets of such therapies should be used to monitor for therapy effects. Large-scale proteomic studies have been performed to help identify such non-targeted protein blood biomarkers. Moreover, as seen with genomic and metabolomic studies, unbiased proteomic studies can be leveraged to quantify AD risk. Proteomic studies performed in blood are motivated by promising results found using CSF and postmortem brain tissue [196,197] as well as studies showing that midlife proteomic signatures predicted all-cause dementia diagnosed 15 years later [198].

Meta-analyses of proteomic studies have identified various proteins associated with AD, compared to cognitively unimpaired individuals, but none of these proteins were individually able to effectively differentiate between MCI and AD or predict conversion from MCI to AD [199,200,201]. Interestingly, a panel of seven proteins was able to predict the AT(N) status (indicated by the presence of Aβ, tau or other markers of neurodegeneration) of European individuals when combined with age and APOE ε4 status (AUC = 0.83 [0.82–0.84]) compared to a model using only age and APOE ε4 status (*p* < 0.05) [202]. In comparing proteomic panels to conventional AD blood biomarkers (Aβ, p-tau181 and NfL), a 19-protein panel is more accurate in diagnosing AD compared to cognitively unimpaired individuals (AUC = 0.98 for protein panel and 0.87 for traditional blood biomarkers, *p* < 0.001) in Hong Kong Chinese individuals [203]. Furthermore, the 19-protein panel correlates with AD progression as measured by plasma p-tau181 and cognitive scores. In a separate Swedish BioFINDER study, a large 74-protein panel classified AD from cognitively unimpaired individuals (AUC = 0.94 [0.87–0.98]) and a 53-protein panel predicted MCI among Aβ-positive individuals (AUC = 0.78 [0.68–0.87]) but did not predict MCI in Aβ-negative individuals, suggesting a unique proteomic profile related to both Aβ deposition and cognitive impairment [204].

In a large proteomic analysis of 931 plasma proteins from 105 individuals with AD and 254 healthy controls, 26 proteins were associated with AD with 16 of them also present in the brain and the CSF proteome; of these, 9 proteins have been replicated in external datasets [205]. Incorporation of these 9 proteins (Table 4) in a prediction model including age and sex yielded an AUC of 0.79, similar to a model using CSF p-tau/Aβ42/40 ratio (AUC = 0.82, *p* > 0.05). Interestingly, a prediction model based on 21 plasma proteins was able to accurately distinguish individuals with AD and TREM2 genetic variants from non-TREM2 AD cases (AUC = 0.90), which was considerably more powerful than using the CSF p-tau/Aβ42/40 ratio (AUC = 0.63, *p* = 1.5 × 10^−4^). This study illustrates that proteomic studies which do not include conventional AD protein biomarkers can accurately predict AD as well as identify individuals who may be harboring high-risk genetic variants, such as those affecting TREM2.

Compared to the more conventional protein blood biomarkers including Aβ and p-tau, knowledge of the temporal changes seen in the AD continuum using blood proteomics is relatively nascent. Nonetheless, there is evidence that such temporal profiles can be developed to help predict AD progression based on findings from the CSF proteome. In the CSF proteome, extracellular matrix-associated proteins and synaptic proteins steadily become abnormal throughout disease progression while levels of proteins involved in glycolytic metabolism and stress responses are more dynamic [206]. Whether a similar temporal profile is also seen in the blood proteome is yet to be determined but may be important in capturing the dynamic changes along the AD continuum.

As with metabolomic studies, there are also limitations with proteomic studies. First, they both utilise mass spectrometry, resulting in the potential for heterogenous results given the variability inherent in this type of technology. Second, as the measurements typically result in relative abundance values, rather than true quantification, it is difficult to interpret values across different studies. Consequently, defining cut-off values for abnormal levels of metabolites and proteins is not straightforward. Finally, as both metabolomic and proteomic approaches are untargeted, it may be difficult to find biological plausibility for certain results. Moreover, findings may be heavily influenced by confounders (both known and unknown), thereby requiring extensive validation studies to increase confidence in and transferability of the results.

## 6. Integrating Blood-Based Biomarkers in Memory Clinics

Establishing the utility of blood biomarkers in ‘real world’ clinical practice is necessary to improve patient care. The appropriate use guidelines for blood biomarkers such plasma p-tau, Aβ, NfL and GFAP are available [207] and are being updated based on recent findings [3]. Currently, it is recommended these biomarkers should only be used in symptomatic individuals with follow-up confirmatory testing using CSF or PET studies. Further studies are needed to help refine the appropriate use of these blood biomarkers without the need of follow-up invasive and costly testing, especially as a panel of markers for patients with undifferentiated symptoms visiting memory clinics.

To this effect, in a pilot study of nearly 30 participants visiting memory clinics in the UK, the average duration from blood collection to plasma p-tau181 results was 3 months, indicating a reasonable turnaround time [208]. Moreover, the p-tau181 result was understandable for 93% of clinicians and it informed AD diagnosis in 44% of cases. In a separate study of 51 patients attending a specialized Thai memory clinic, plasma p-tau181 collection and analysis was less expensive and just as accurate as an Aβ-PET scan (AUC = 0.84 [0.73–0.94]), indicating a more feasible means to measure biomarkers in this community [119]. Similarly, in a prospective study of over 350 patients attending a Spanish memory clinic, plasma p-tau181 demonstrated Class I evidence for correlation with AD [118]. Moreover, by taking a multi-omics approach and incorporating an AD-PRS with plasma p-tau181, prediction of Aβ-PET status improved from 68% (for p-tau181 alone) to 88% (*p* = 0.001) [209]. Similarly, the prediction accuracy of the underlying cause of cognitive impairment in individuals from specialized memory clinics followed over a 4-year period was higher with a diagnostic model combining p-tau217 or p-tau181 with cognitive tests and APOE status (AUC 0.90 [0.86–0.94]), compared to predictions made by memory clinic specialists (AUC 0.72 [0.65–0.78], *p* < 0.001) [210]. These studies indicate that incorporating blood biomarkers into the clinical work-up of patients with cognitive symptoms is feasible and improves diagnostic accuracy.

Unlike participants in research studies, patients in memory clinics are more diverse with a wider range of co-morbidities and ethnicities. Studies have shown an increase in plasma biomarkers levels in cognitively unimpaired individuals with various medical conditions [211,212,213]. For example, in a large community-based cohort of over 1300 participants, levels of plasma p-tau181 and p-tau217 were associated with body mass index, chronic kidney disease, hypertension and prior stroke and myocardial infarction, even when levels were adjusted for age, sex and Aβ status [214]. Plasma NfL and GFAP levels also correlated with renal function [215,216]. Interestingly, while individual biomarkers can be affected by renal function, use of ratios, in particular that of p-tau217 to total-tau217, is less sensitive to renal impairment and correlates with AD [217]. Overall, co-morbidities, especially renal impairment, affect how biomarker values are interpreted for AD diagnosis [218]. Another variable in clinic patients is their fasting status with studies showing that blood biomarkers are strongly affected by the postprandial status [219,220].

The influence of ancestral background on protein blood biomarkers is not clear yet. Plasma Aβ42/40 ratio is more robust than p-tau181, NfL and p-tau231 when comparing AUC values for association with brain Aβ levels in African Americans, different than what is reported in European populations [221]. Some studies have found that levels of these biomarkers differ significantly amongst African Americans, non-White Hispanics and Latino Americans while others found non-significant differences [86,222,223].

It is clear that variabilities inherent to patients in memory clinics, compared to participants in research studies, can significantly impact blood biomarker results. Further research with a diverse population is desperately needed to clarify preliminary findings as well as determine clinically appropriate cut-off values for blood biomarkers based on confounders such as co-morbidities, ethnicity and fasting status. There is also an equally important and outstanding question about which assays for blood biomarkers are most robust in the clinical settings. Many studies with head-to-head comparisons between different assays, including retest variability, have recently been published [169,224,225,226,227,228], and offer insights into this question. However, a consensus testing method has not yet been established.

## 7. Discussion and Future Directions

There has been an explosion of studies delineating blood biomarkers in AD, largely driven by improved technology, lower costs and increased access. Genomic, proteomic and metabolomic biomarkers have all demonstrated the ability to characterize AD risk and discriminate from other causes of dementia with a high degree of accuracy within the tested populations. Furthermore, each type of ‘omic’ data contributes uniquely to the AD risk with PRS capturing the static and baseline risk, while proteomics and metabolomics represent a more dynamic risk profile along the AD continuum. With this ‘multi-omic’ risk stratification approach, individuals can theoretically be stratified initially along their genetic risk for AD, with high-risk individuals following a different paradigm of clinical and biomarker assessments than those with low genetic risk. However, prospective cohort studies are needed to evaluate and validate this model of a personalized diagnostic pathway for AD diagnosis and risk stratification. In particular, testing these biomarkers in more diverse populations, especially with respect to ethnicity and co-morbidities, is crucial to widely implementing biomarker-based assessments in memory clinics. Early results in this area are promising and provide hope for an integrated ‘multi-omics’ approach to AD risk prediction.

## 8. Search Methodology

PubMed was searched for articles, including advanced online publications published up until 28 September 2023, using the following search terms “Alzheimer’s disease”, “genome wide association studies”, “polygenic risk scores”, “blood biomarkers”, “p-tau”, “amyloid beta”, “neurofilament light chain”, “glial fibrillary acidic protein”, “YKL-40”, “TREM2”, “proteome”, “metabolome” and “memory clinic”. Full versions of review and original articles were assessed for appropriateness. References were also examined to identify additional articles not identified using the search terms listed.

## Figures and Tables

**Figure 1 ijms-25-01231-f001:**
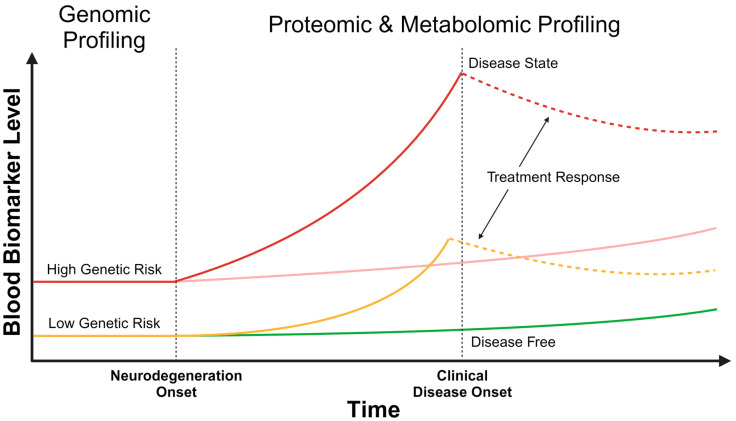
Temporal changes in multi-omic blood biomarkers. At birth, an individual may have a high (red) or low (yellow) static genetic risk for Alzheimer’s disease, which can be identified by genomic profiling. After the onset of neurodegeneration, the dynamic risk for developing Alzheimer’s disease can be assessed with proteomic and metabolomic profiles. Changes in blood biomarker levels may be detected well before clinical disease onset, as shown with the rising solid red and yellow curves. Proteomic and metabolomic levels can be reassessed over time, with assessment frequency based on the static genetic risk, to understand the trajectory of biomarker change and to develop a contemporaneous risk prediction. This will help identify those individuals in whom the blood biomarkers are rising quickly (solid red and yellow curves), compared to others in whom the change is more gradual (pink and green lines). With the onset of clinical disease (solid red curve), and potentially prior to the onset (solid yellow curve), treatment can be initiated, altering proteomic and metabolomic blood biomarker levels (dashed red and yellow curves). Depicted with the dashed lines is a treatment response that is reducing blood biomarker levels.

**Table 1 ijms-25-01231-t001:** Characteristics of recent Alzheimer’s disease genome-wide association studies [21].

Study	Number of Cases	Number of Controls	Percentage of Proxy Cases	Total Significant Loci Identified	Novel Significant Loci Identified
Bellenguez et al., 2022 [17]	111,326	677,663	42%	75	42
Wightman et al., 2021 [18]	90,338	1,036,225	52%	38	7
de Rojas et al., 2021 [19]	97,796	369,827	43%	35	6
Jansen et al., 2019 [22]	71,880	383,378	65%	29	13
Kunkle et al., 2019 [23]	34,274	59,163	0%	25	5

**Table 2 ijms-25-01231-t002:** Illustrative performance comparisons of blood protein biomarkers for classifying Alzheimer’s disease from recent large studies.

Disease Comparison	Study	Blood Biomarker Performance (AUC [95% Confidence Interval Range])					
Preclinical AD vs. CU	Chatterjee et al., 2022 [167] (*n* = 95)	**GFAP** **0.79 (0.69–0.89)**	NfL 0.61 (0.47–0.74)	t-tau 0.61 (0.48–0.75)		p-tau181 0.74 (0.63–0.85)	p-tau231 0.77 (0.68–0.87)
		*p* < 0.005		*p* < 0.05 vs. GFAP		Not significant vs. GFAP	
AD vs. CU	Simren et al., 2021 [158] (*n* = 202)	**p-tau181** **0.91 (0.86–0.96)**	NfL 0.82 (0.79–0.92)	GFAP 0.69 (0.57–0.77)		t-tau 0.70 (0.61–0.79)	Aβ_42/40_ 0.67 (0.58–0.76)
		*p* < 0.001		Not significant vs. p-tau181			
MCI Conversion to AD vs. Non-Conversion	Simren et al., 2021 [158] (*n* = 107)	**p-tau181** **0.77 (0.61–0.84)**	Aβ_42/40_ 0.67 (0.51–0.82)	NfL 0.62 (0.45–0.78)		t-tau 0.60 (0.42–0.79)	GFAP 0.61 (0.54–0.72)
		*p* < 0.05		Not significant vs. p-tau181			
AD vs. FTD	Thijssen et al., 2021 [124] (*n* = 349)	**p-tau217** **0.93 (0.91–0.96)**			p-tau181 0.91 (0.88–0.94)		
		*p* = 0.01					
AD vs. non-AD Dementia	Palmqvist et al., 2020 [123] (*n* = 81)	**p-tau217** **0.89 (0.81–0.97)**			p-tau181 0.72 (0.60–0.84)		
		*p* = 0.04					

Bolded blood protein biomarkers are those significant compared to biomarkers in grey boxes. AD, Alzheimer’s disease; CU, cognitively unimpaired; MCI, mild cognitive impairment; FTD, frontotemporal dementia; AUC, area under the ROC curve; GFAP, glial fibrillary acidic protein; NfL, neurofilament light chain; t-tau, total tau; p-tau, phosphor-tau; Aβ, amyloid-β.

**Table 3 ijms-25-01231-t003:** List of 11 replicated metabolites associated with Alzheimer’s disease compared to cognitively unimpaired individuals [184].

Direction Change	Metabolite
Increased level associated with higher risk	Glycerophosphocholine
	Aspartic acid
	Hydroxypalmitic acid
	Choline
Decreased level associated with higher risk	Hexanoylcarnitine AcCa (6:0)
	4-Decenoylcarnitine AcCa (10:1)
	Tetradecadiencarnitine AcCa (14:2)
	Piperine
	Decanoylcarnitine AcCa (10:0)
	L-Acetylcarnitine
	Serotonin

AcCa, acylcarnitine.

**Table 4 ijms-25-01231-t004:** List of 9 replicated plasma proteins predictive of Alzheimer’s disease [205].

Protein	Full Name	*p*-Value AD vs. CU
**ERK-1**	Mitogen-activated protein kinase 3	1.79 × 10^−9^
**BARK1**	Beta-adrenergic receptor kinase 1	2.13 × 10^−8^
**GNS**	N-acetylglucosamine-6-sulfatase	4.10 × 10^−7^
**CAMK2D**	Calcium/calmodulin-dependent protein kinase type II subunit delta	7.65 × 10^−7^
**CDON**	Cell adhesion molecule-related/down-regulated by oncogenes	2.25 × 10^−6^
**HMG-1**	High mobility group protein B1	6.76 × 10^−6^
**tPA**	Tissue-type plasminogen activator	1.22 × 10^−5^
**RELT**	Tumor necrosis factor receptor superfamily member 19L	1.94 × 10^−5^
**Integrin α1β1**	Integrin alpha-I: beta-1 complex	1.44 × 10^−4^

AD, Alzheimer’s disease; CU, cognitively unimpaired.

## Data Availability

No new data were created or analyzed in this study. Data sharing is not applicable to this article.

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
