# Peer review of "Multi-Omic Blood Biomarkers as Dynamic Risk Predictors in Late-Onset Alzheimer’s Disease"

_ijms, 2024, doi:10.3390/ijms25021231_

Round 1

Reviewer 1 Report

Comments and Suggestions for Authors

1. Overall comment thought out the manuscript incorrect reference style for IJMS.

2. Line 72: incorrect symbol for A_

3. Line 143, 355, 467: Figure number is missing

Questions to address:

1) Overall, the manuscript nicely combines recent findings and incorporates monoclonal therapy results, yet there are literature data for the low molecular weight compounds: Galantamine, Rivastigmine, Donepezil, and Memantine (DOI: 10.1002/med.21434) which are widely used to treat symptoms in mild to moderate AD. None of them delay decline in memory and/or reduce symptoms, and unable to stop the disease from worsening over time. However, a section on these treatments would greatly benefit the scope of the manuscript. A quick pubmed search revealed that there are several studies that should be discussed here as well.

Reviewer 2 Report

Comments and Suggestions for Authors

Multi-omic Blood Biomarkers as Dynamic Risk Predictors in Late-Onset Alzheimer’s Disease

Authors provide a thorough review on peer-reviewed studies that develop risk scores and search for predictive biomarkers of late-onset Alzheimer’s Disease.   Authors focus on the use of omic technologies, specifically genomic (GWAS), proteomic, and metabolomic, and provide details on results from seminal studies in the field. Overall, the manuscript is well written and organized, and provides a lot of details on this topic.  While authors provide some key takeaway points throughout the manuscript, these points could be summarized more clearly in the conclusion, providing the readership with a more clear high level view of where the field has advanced in the recent years as well as recommendations from the authors on what the field should focus on next to increase adoptability of biomarkers for AD in the clinic.

Major Comments:

·      In general, authors present on results from numerous studies that evaluated GWAS, proteomic, and metabolomic profiling in the context of AD.  From the results, it is unclear whether or how results from these studies were validated, or whether results were consistent across studies (including across studies performing meta-analyses).  The readership would benefit from a summary of what types of validation have been or should be performed to advance the ability of using such omic profiling in the diagnosis or risk assessment of AD.

·      Tables 2 and 3 report on results for a single study.  Why were those specific studies chosen to highlight in a table format?  In this age of AI, results from these studies may be more readily picked up than those highlighted in the text.  Authors should clarify why these studies were chosen as ones with more visibility/prominence. 

·      Generally, there are conflicting results regarding the optimal blood protein biomarkers that predict AD. Authors posit that these conflicting results may be due to nuances in disease stages that were assessed (p. 13, lines 345-346) or perhaps these could be due to different populations being studied. Could authors clarify the disease stages and populations or other factors a bit more?  Can authors clarify what is meant by “a more nuanced approach depending on disease stage will be required”?

·      Table 1 highlights AD GWAS studies within the last 4 years.

o   Could authors clarify whether the last column (“Novel GWAS Loci (from prior studies”) is the number of loci that are unique to the study in question compared to all other studies in the table?  Are there any loci that overlap across all these studies?

o   Also, could authors clarify whether there is any overlap between the studies used for meta-analyses in these studies? 

o   Lastly, basic demographics (e.g., race/ethnicity, sex, etc.) are missing and should be included.

·      Regarding Figure 1:

o   Generally, the figure legend could provide more information on how to interpret this summary figure.  It is important to include the key takeaway points here.

o   Could authors clarify whether the red line that doesn’t spike up at disease onset depicts disease free individuals?

o   Grammar issue in “and can be based the underlying genetic risk”.

o   A key point in this figure, which is not made explicit in the figure legend or text, is the importance of evaluating biomarker levels over time and using an individual’s levels assessed earlier in time as a baseline.

o   Another key point mentioned by authors is how metabolomic and proteomic profiles complement genomic profiling in that they can pick up changes closer to disease onset (before and after).  This could be clarified in the legend.

o   Another key point mentioned by authors is that while treatments show a marked decrease in potential biomarkers (e.g., p-tau217, p-tau181), how these markers change after treatment has been discontinued is unknown.  Is there a way to show this more clearly in the figure?

o   Consider changing the y-axis label “Blood Biomarker Levels” to “Changes in Blood Biomarkers”.

o   In the figure legend, authors state “Therapy can reduce levels of proteomic and metabolomic blood biomarkers”.  However, couldn’t levels of proteins and metabolite biomarkers be augmented by therapy?

·      Regarding Table 2:

o   Authors could show the number of individuals assessed in these studies, as well as basic demographics (e.g., race/ethnicity, sex, etc.).

o   How are the p-values calculated (e.g., t-tests)? 

o   For the second study, Simren et al, it’s unclear why p-tau181 is not significant compared to GFAP/t-tau/Aβ42/40 given the AUC[CI] listed.  Similarly, for the Thijssen et al. study, it is surprising that a significant difference was found.

o   The location of the bold vertical lines in the table are confusing since different proteins were assessed in different studies.

o   Reference #164 is mentioned in the text (first paragraph of P.13), not in Table 2, and shows a result contrasting that found in reference #158 which is present in Table 2.  Why not also show #164 in Table 2? 

o   The studies are also difficult to compare in this table because the ‘control’ group changes from study to study.  Perhaps providing a timeline of advancement to AD and mapping the biomarkers found along that timeline could help?

·      Regarding Table 3:

o   Authors could show the number of individuals assessed in these studies, as well as basic demographics (e.g., race/ethnicity, sex, etc.).

o   Authors could clarify whether the confidence in the identity of the metabolites (e.g., level 1 vs. levels 2 or 3).

·      P. 15, lines 384-389: Authors highlight the fact that there is heterogeneity in results observed in metabolomic studies due to mass spectrometry technologies.  However, these limitations are not highlighted as part of the proteomic studies yet many of those rely on mass spectrometry technologies. 

o   Authors should clarify limitations of proteomic studies as well as metabolomic studies, and mass spectrometry untargeted data in general. 

o   It’s important to state that many omic studies produce relative abundance values rather than true quantitation, making it more difficult to define consistent cutoffs across studies.

·      For studies that evaluate metabolites and their associations with AD risk, are there any overlap in metabolites found to be discriminating?  Why was study reference 181 shown in Table 3 and not others?  Similarly, why does Table 4 focus on a single proteomic study and not show others?

·      In the Discussion/Conclusions, authors state “Incorporating these different approaches into a ‘multi-omics’ step-wise approach may offer a more feasible means for clinical applicability and understanding AD risk.”  Could authors elaborate on this point?  How is this feasible given that the authors state, in the previous sentence, that each ‘omic’ approach requires a high level of expertise and is costly?

Minor Comments:

·      P. 4, line 100: typo, ‘enrol’ should read ‘enroll’.

·      Authors are encouraged to include the number of study participants for all studies that are mentioned throughout the manuscript. 

·      P. 13, line 320: type at “AUC = ¬¬¬0.67-“.

·      P.18, line 473: authors state that omic approaches are costly.  Could a range be provided?  How do these compare to conducting neuropsychological and other tests?   

·      P.9, line 245: authors state that biomarkers could refine “understanding of the underlying pathophysiology” yet they do not discuss pathophysiology in the paragraph.

Comments on the Quality of English Language

There are only a few typos detected and are included in comments above.

Reviewer 3 Report

Comments and Suggestions for Authors

Ref: ijms-2786941

The authors present a narrative review concerning blood biomarkers as risk predictors in late- onset Alzheimer’s disease.

 Blood-based biomarkers offer certain advantages compared to PETs and CSF-based biomarkers, since they are less invasive, require no radiation and have a lower cost. Thus, they have received much attention recently, especially in the era of anti-amyloid treatments.

 The review is very comprehensive, educative and covers practically all aspects of blood basted multi-omics, related to Alzheimer’s disease. From the genetic point of view, the authors present current data not only on the role of APOE, but they also discuss the polygenic risk score. As regards the role of biomarkers related to neurodegenerative processes, they discuss the advantages and the drawbacks of Aβ42 or Aβ42/40 ratio, they share with us current knowledge concerning the various types of p-tau (181, 217, 231) and even the role of inflammation-related or other proteins. Finally they present knowledge and suggestions concerning the integration of these biomarkers in memory clinics, for possible use in every-day practice and for possible improvement of diagnostic accuracy throughout the Alzheimer’s continuum.

I was excited by this review. When and if published, I will suggest that to my colleagues and I intent to cite it in my own submissions. 
